# Neurobiological Signatures of Auditory False Perception and Phantom Perception as a Consequence of Sensory Prediction Errors

**DOI:** 10.3390/biology11101501

**Published:** 2022-10-13

**Authors:** Min-Hee Ahn, Nour Alsabbagh, Hyo-Jeong Lee, Hyung-Jong Kim, Myung-Hun Jung, Sung-Kwang Hong

**Affiliations:** 1Laboratory of Brain & Cognitive Sciences for Convergence Medicine, Hallym University College of Medicine, Anyang 14068, Korea; 2Department of Otorhinolaryngology-Head and Neck Surgery, Hallym University College of Medicine, Anyang 14068, Korea; 3Department of Psychiatry, Hallym University College of Medicine, Anyang 14068, Korea

**Keywords:** efference copy, auditory perception, Bayesian brain, tinnitus, auditory hallucination

## Abstract

**Simple Summary:**

The principle of Bayesian inference provides a theoretical framework for stable perception in numerous tasks, including sensory–perceptual tasks, sensorimotor, and motor tasks. Efference copy (EC) signals enable organisms to reduce cognitive loading by decreasing the sensory processing of their own actions. In the auditory domain, the sensorimotor prediction error is responsible for false perceptions, such as auditory hallucinations in schizophrenia, while sensory prediction error also leads to auditory phantom perception such as tinnitus. Attenuation of the N1 component in event-related potentials has been suggested as evidence for the integrity of the EC mechanism. The current study investigated the EC mechanism in auditory false perception and phantom perception as a consequence of sensory prediction errors. N1 attenuation failures were present in tinnitus patients with significant hearing impairment and in those with schizophrenia, indicating that phantom perception after sensory deprivation might lead to impairment in the EC mechanism. However, the corresponding neural representation with spatiotemporal patterns presented differently between patients with schizophrenia and tinnitus. Although the present study had several constraints, the results provide a new perspective on neurobiological aspects of abnormal auditory perception resulting from deficits in predictive coding.

**Abstract:**

In this study, we hypothesized that top-down sensory prediction error due to peripheral hearing loss might influence sensorimotor integration using the efference copy (EC) signals as functional connections between auditory and motor brain areas. Using neurophysiological methods, we demonstrated that the auditory responses to self-generated sound were not suppressed in a group of patients with tinnitus accompanied by significant hearing impairment and in a schizophrenia group. However, the response was attenuated in a group with tinnitus accompanied by mild hearing impairment, similar to a healthy control group. The bias of attentional networks to self-generated sound was also observed in the subjects with tinnitus with significant hearing impairment compared to those with mild hearing impairment and healthy subjects, but it did not reach the notable disintegration found in those in the schizophrenia group. Even though the present study had significant constraints in that we did not include hearing loss subjects without tinnitus, these results might suggest that auditory deafferentation (hearing loss) may influence sensorimotor integration process using EC signals. However, the impaired sensorimotor integration in subjects with tinnitus with significant hearing impairment may have resulted from aberrant auditory signals due to sensory loss, not fundamental deficits in the reafference system, as the auditory attention network to self-generated sound is relatively well preserved in these subjects.

## 1. Introduction

When adding new information, the brain can compute updated probabilities based on prior knowledge and experience. Thus, Bayesian inference operates as a prediction system that selectively uses past experiences to predict upcoming events, and thus minimizes uncertainty [1]. As many of our actions are accompanied by auditory consequences, such as speaking, hearing our own voice, or playing instruments, sensorimotor learning (integration) is a prerequisite for the stable auditory perception of the upcoming state in the dynamical aspect of Bayesian inference, which can be achieved by updating recent efference copy (EC) signals as a representation of the consequences of the action. For example, the copy signal of the motor command is forwarded to the corresponding sensory cortex during the preparation of motor action execution [2,3,4]. Subsequently, the actual sensory consequences of the motor command are compared with the copy signal in the corresponding sensory cortex, where the brain determines whether the sensory reafference from motor outputs is self-generated [4,5].

Attenuated auditory cortical responses during self-vocalization reflect suppression of the auditory response mediated by the EC conveyed to the auditory cortex, in which attenuation of the N1 component in event-related potentials (ERPs) has been suggested as evidence for the integrity of the EC mechanism [6,7]. In contrast, sensory attenuation failures indicate deficits in the EC mechanism for sensorimotor integration, which has been reported to be associated with false auditory perceptions, such as auditory hallucinations in schizophrenia [4,6,8,9]. Interestingly, studies have attributed phantom perceptions, such as tinnitus, to top-down prediction errors by the change in incoming auditory input after auditory deafferentation [9,10,11,12,13,14]. Perceptual phenomena in tinnitus can also be modeled based on the Bayesian brain hypothesis [1].

We focused on neuroscientific evidence that there may be a cerebellum-like circuit that cancels responses to self-generated sounds at the earliest level (brainstem) for auditory processing [15]. The dorsal cochlear nucleus (DCN) operates as the key structure at the earliest level [16], and hearing deafferentation leads to a decrease in auditory input with a subsequent increase in somatosensory input to the DCN, which may play a role in the pathogenesis of tinnitus [17]. Moreover, the auditory predictions at higher stages in the hierarchy of auditory processing are made in the paraflocculus in the cerebellum, which is thought to integrate ECs [18]. Based on those studies, we hypothesized that impairment of the EC mechanism would be marked in subjects with tinnitus with significant hearing deficits as modulation of efferent signals can occur due to pronounced auditory deafferentation.

To test our hypothesis, we used the “Talk/Listen” paradigm coupled with electroencephalography (EEG) to explore EC mechanisms in patients with tinnitus (phantom perception) and auditory hallucinations. We divided the tinnitus patients into two subgroups based on a self-reported visual analog scale (VAS) score: tinnitus without hearing impairment (T1) and tinnitus with hearing impairment (T2). We then investigated the spatiotemporal representation of auditory perception to self-generated sound in these subjects in comparison to healthy individuals.

## 2. Methods

### 2.1. Participants

This study included 23 patients with tinnitus (8 women, mean age: 39.95 ± 12.00 years), 10 schizophrenia patients (7 women, mean age: 30.10 ± 6.00 years), and 23 healthy volunteers (12 women, mean age: 34.43 ± 7.14 years). All participants were right-handed. This study was conducted in accordance with the ethics guidelines established by the Institutional Review Board of Hallym University (approval #: 2018-10-021-004). For the healthy and schizophrenia groups, individuals were excluded if they had any significant head injury (e.g., traumatic brain injury), cognitive disabilities, other medical or neurological history affecting the central nervous system, and/or prior history of otologic symptoms (e.g., tinnitus, hyperacusis, hearing difficulty, dizziness, and noise exposure). Schizophrenia was diagnosed using the Diagnostic and Statistical Manual of Mental Disorders IV [19] by a certified psychiatrist (Jung, M-H). The mean duration of schizophrenia was 7.50 ± 4.06 years (range: 2–16 years). In addition, the schizophrenia symptoms were assessed using the Positive and Negative Syndrome Scale (PANSS) [20], which showed an average score of 52.7 ± 14.23. The average value of hallucinatory behavior (P3) in PANSS was 2.30 ± 1.25 and 6 patients experienced auditory hallucinations during the present study. All participants in the schizophrenia group were receiving neuroleptic medications at the time of the experiment. The hearing levels were tested at a conventional frequency range of 0.25–8 kHz with calibrated pure tone audiometry (GSI AudioStar ProTM, Grason Stadler, Eden Prairie, MN, USA) in a soundproof audio booth. Both schizophrenia and the healthy group showed normal hearing levels on a pure tone audiogram (Table 1).

For the tinnitus group, participants were excluded if they had a history of dizziness, ototoxic drug use, poorly defined or complex tinnitus on a tinnitus matching test (failure of tinnitus pitch matching), and/or active or prior history of psychiatric disorders. Tinnitus patients underwent extended high-frequency audiometry (0.25–12 kHz). Tinnitus patients completed a tinnitus questionnaire including a Korean version of the Tinnitus Handicap Inventory (K-THI) questionnaire, translated from the original THI [21]. The hearing difficulty associated with tinnitus was assessed by VAS scores ranging from 1 to 5 (1: no discomfort; 2: mild hearing impairment; 3: moderate hearing impairment; 4: severe hearing impairment; 5: profound hearing impairment). We hypothesized that substantial sensory prediction error may influence sensorimotor integration using EC signals. However, we could not quantitatively define the hearing level leading to significant sensory prediction error because all subjects with tinnitus may have hidden hearing loss, which cannot be detected with conventional hearing tests [22]. Furthermore, the perceptual consequences following auditory deafferentation are complex and go far beyond the reduced sensitivity demonstrated by a conventional audiogram [23]. Therefore, although we investigated objective hearing level including high-frequency range in subjects with tinnitus, the tinnitus patients were subdivided into two groups based on the VAS to focus on perceptual hearing discomfort; the T1 group consisted of patients without hearing impairment (VAS grades 1 and 2) accompanied by tinnitus, while the T2 group included patients with hearing impairment accompanied with tinnitus (VAS grades 3–5).

The pure tone average (PTA, average of hearing threshold levels at 0.5 kHz, 1 kHz, 2 kHz, and 3 kHz) [24] and high-frequency PTA (average of hearing threshold levels at 8 kHz and 12 kHz) of the involved tinnitus side, as well as tinnitus loudness (if both ears had tinnitus, the more dominant side was assessed) were obtained from participants in the tinnitus group. An alternative forced choice method was employed to measure tinnitus pitch in tinnitus subjects [25]. Pulsed pairs of tone ranging from 125 Hz to 12 kHz were alternatively presented at a comfortable hearing level (10–15 dB above the subject’s hearing threshold) to the involved side. Subsequently, subjects with tinnitus were asked to determine which one most closely matched their tinnitus. These procedures were repeated seven to nine times for a correct match. After patients chose which stimulus was closest to the pitch of their tinnitus, the same two alternative forced choice procedure was repeated with the tinnitus matched tone and the octave above and below it to avoid octave confusion [14]. If patients described their tinnitus as hissing, swishing, or pulsatile, these subjects are excluded from the final analysis as it is hard to match tinnitus loudness (pitch match failure). The loudness of the matching sound was determined when subjects expressed a synchronization of tinnitus intensity in the same frequency in pitch matching. If the patient exhibited differences in tinnitus between the two ears, the tinnitus pitch was decided based on the side considered more annoying by the patient. The T2 group showed a PTA of 29.1 ± 19.16 dB, high-frequency PTA of 65.38 ± 26.6 dB, and loudness of 63.08 ± 20.35 dB; these values in the T1 group were 14.5 ± 10.08 dB, 31.35 ± 22.07 dB, and 31.28 ± 20.46 dB, respectively. Notably, the T2 group had a significantly higher hearing threshold (*p* = 0.026 for PTA, *p* = 0.006 for high-frequency PTA, and *p* = 0.004 for loudness; Table 1, Figure 1A,B). EEG data were obtained from 56 participants; 2 patients with tinnitus were excluded from the current analysis because of poor data quality.

### 2.2. Equipment Setup

The experiment was performed in a soundproof room with each participant seated upright in a comfortable chair facing a monitor where the experimental stimuli were presented. A recording microphone was placed 3–5 cm in front of the participant’s mouth and binaural insert earphones (EARTONE 3A^®^, 3M Company, Indianapolis, IN, USA) were used for sound stimuli delivery. Two separate computers were installed outside of the participant testing room for EEG recording and stimuli presentation, respectively. The stimulus presentation computer was connected to the microphone to record the participant’s utterances (during the Talk condition), which were later delivered to the participants using the insert earphones. In addition, the stimulus presentation computer was linked to the EEG recording computer to enable the insertion of stimulus-locked triggers in the EEG data.

### 2.3. Talk/Listen Paradigm

The experimental paradigm was adapted from the work of Ford and colleagues [6]. The task paradigm was composed of two consecutive conditions: Talk and Listen. Each condition consisted of 25 blocks with 8–10 stimuli per block (200–250 stimuli per condition). In the Talk condition, participants were required to produce the sound “ah” 8–10 times per block, sharply and briskly, with an interval of 2–3 s between utterances. In the Listen condition, participants were asked to listen to their own recorded “ah” utterances. The entire test paradigm was performed using E-Prime software (Psychology Software Tools, Pittsburgh, PA, USA).

Before starting the actual test, a practice session was conducted for both the Talk and Listen conditions in which the participants watched an instructiona movie that demonstrated how the Talk/Listen paradigm was performed. For the Talk condition, participants were trained to correctly produce the sound “ah” during the Talk condition while maintaining the same sound loudness level of 75–85 dB SPL using a sound level meter (Type 2250, Brüel & Kjær Sound and Vibration Measurement, Naerum, Denmark) attached to the microphone; each participant’s vision was fixated on the cross “+” sign on the monitor.

During the Listen condition, the participant’s recordings were played back to them using E-prime 2.0 software (PST Inc., Pittsburgh, PA, USA). Participants were trained to fixate their vision on the cross “+” sign while listening to playback of their utterances.

Participants were also asked to minimize mouth movements, eye blinks, and any bodily movement when producing “ah” utterances (during the Talk condition) and when listening to their recordings (during the Listen condition) to avoid the inclusion of artifacts in the EEG data. Participants were instructed to raise one of their hands at the beginning of each block to inform the examiner that they were ready. Whenever a participant raised their hand, the cross sign “+” was presented in the middle of the monitor, indicating that the participant should begin producing an “ah” sound (during the Talk condition) or listen to their recording (during the Listen condition), while fixating their vision on the cross sign until the “rest” sign appeared; this rest sign allowed the participant to take a break and prepare for the next block. Time trigger markers were inserted into the EEG data using E-Prime software whenever the cross sign appeared on the screen (i.e., when the participant started a test block). The onset of “ah” utterances was detected manually for each test block using Adobe Audition CS6 (Adobe Systems Incorporated, San Jose, CA, USA); these onsets were later added to the time trigger marker for further EEG analyses.

### 2.4. EEG Data Acquisition and Processing

The EEG recording was performed separately for each condition, and the actual recording required approximately 25 min for the Talk condition and 20 min for the Listen condition. The EEG was recorded using the BrainAmp DC amplifier with a 64-channel actiCAP (Brain Products, Munich, Germany), corresponding to the international 10/20 system of electrode placement, with a sampling rate of 1000 Hz. Eye movement activity was monitored using an electro-oculogram (EOG) electrode placed suborbitally onto the left eye; vertical and horizontal electro-ocular activities were computed using two pairs of electrodes attached vertically and horizontally to both eyes (i.e., Fp1 and EOG for the vertical EOG; F7 and F8 for the horizontal EOG). EOG artifacts were corrected offline using independent component analysis methods [26]. For the ERP study, EEG data were epoched from 400 ms pre-stimulus to 1000 ms post-stimulus. Electrode impedance was maintained at <5 kΩ during the recording sessions.

Raw EEG data were pre-processed using Brain Vision Analyzer software (version number 2.2, Brain Products). A 0.5 Hz high-pass filter and 50 Hz low-pass filter were applied to the EEG raw data. Subsequently, EEG data were re-referenced to the common average reference and thoroughly inspected for artifact rejection (e.g., eye blinks, jaw clenching, eye movements, and bodily movements) and interpolation of channels that were not functioning appropriately. ERPs were then analyzed using MATLAB (ver. R2021a, MathWorks, Inc., Natick, MA, USA) and Python (ver. 3.6). For each participant, the ERPs were averaged for the epochs of Talk and Listen conditions separately. Subsequently, the ERPs for all participants within the same group (healthy, T1, T2, and schizophrenia) were averaged separately. The N1 component of ERPs (peak 50–150 ms post-stimulus) was measured separately for the Talk and Listen conditions for each group. The suppression effect of N1 was estimated at the Fz, Cz, and Pz electrodes. All time windows were based on their grand averages while taking individual variations into account. Baseline corrections were performed using the 500–0 ms pre-stimulus interval. The amplitudes and latencies of each peak were compared for ERP analysis

### 2.5. Granger Causality Analysis

Granger causality analysis was performed to explore the potential pathways of auditory perception. This method can provide directional causal interactions among electrophysiological signals from functional brain areas [27]. Functional connectivity was mapped for each experimental condition using eConnectome software, which provides a critical current density model to solve the inverse problem from EEG scalp electrodes [27,28]; the details of this approach were described in our previous study [14,29]. In the present study, Granger causality was investigated at the grand-averaged evoked alpha activity of frequency range between 8 and 13 Hz from −300 ms to 0 ms pre-stimulus to explore the early auditory attentional process. It was then investigated from 0 ms to 250 ms post-stimulus to investigate the auditory attentional control networks. As alpha activity has been suggested to function as a gating mechanism during auditory attention in both the auditory cortex (early) and the frontoparietal attentional control network (late) [30], we analyzed the brain network in the frequency range. Based on the most pronounced cortical activity for auditory attentional processing, as estimated by the eConnectome software, 22 bilateral ROIs were selected (i.e., BA24L/R, BA22L/R, BA27L/R, BA28L/R, BA32L/R, BA34L/R, BA41L/R, BA42L/R, BA44 L/R, BA45L/R, and BA46L/R) to map directional connectivity. Source waveforms were estimated at the 21 ROIs (BA34 L was excluded for focusing auditory perception in the final analysis) and the direct transfer function analysis was used for directional information flow across the sources.

### 2.6. Statistical Analysis

All statistical analyses were performed using MATLAB (ver. R2021a; MathWorks, Natick, MA, USA) or SPSS Statistics (ver. 22; IBM, Armonk, NY, USA). The paired *t* test was used for comparison of N1 amplitude among groups. We adapted nonparametric analysis based on surrogate data to test the significance of the estimated connectivity [31,32]. In this method, after transformation of the initial time series to the Fourier space, the phases were randomly shuffled without changing the magnitude in the Fourier space. The surrogate data in the Fourier space were then converted back to the time domain. As phase shuffling preserves the spectral structure of the time series data, the calculated original connectivity values were compared to values under a null hypothesis of no significant connectivity using the shuffling process. The shuffling and connectivity estimation procedures were repeated 1000 times in this study. Statistical assessment of connectivity was performed using the surrogate approaches (1000 surrogate datasets) for connectivity analysis and the critical value of significance was set as *p* < 0.05 [31].

## 3. Results

### 3.1. Evidence for the EC Mechanism in Normal Volunteers

The analysis of auditory-ERPs based on conditions (Talk vs. Listen) at the Fz site showed a significant reduction (*t* = 2.643, *p* = 0.023) of N1 amplitude (peak: −0.0699 µV, mean ± standard deviation, SD: −0.853 ± 0.672 µV) in the Talk condition, compared with the Listen condition (peak: −1.846 µV, mean ± SD: −1.964 ± 1.247 µV); this was reflective of the functional integrity (inhibition) of the EC mechanism. Similarly, at the Cz site, the N1 amplitude was significantly suppressed (*t* = 5.503, *p <* 0.001) in the Talk condition (peak: −0.012 µV, mean ± SD: −0.096 ± 0.852 µV) compared with the Listen condition (peak: −1.538 µV, mean ± SD: −1.594 ± 0.880 µV; Figure 1C). Conversely, the N1 amplitude at the Pz site did not significantly differ between the Talk and Listen conditions (*t* = −0.529, *p* = 0.607). In summary, the effect of the EC mechanism in suppressing self-generated sound was more prominent in the frontocentral areas (Fz and Cz) of the brain, which apparently reflects the potential EC pathways.

### 3.2. Evidence for Altered EC Mechanism in the T1 Group vs. T2 Group

Notably, the T1 group showed significant suppression at the Fz site (*t* = 2.750, *p* = 0.028) of N1 amplitude (peak: −0.382 µV, mean ± SD: −0.608 ± 0.405 µV) in the Talk condition, compared with the Listen condition (peak: −1.348 µV, mean ± SD: −1.466 ± 0.896 µV). The significant attenuation of N1 amplitude in the Talk condition (peak: −0.422 µV, mean ± SD: −0.483± 0.88 µV) versus the Listen condition (peak: −1.545 µV, mean ± SD: −1.613 ± 0.946 µV) were consistent with healthy volunteers at the Cz site (*t* = 2.995, *p* < 0.05, Figure 1D). Additionally, attenuation failure on N1 amplitude in the Talk condition was consistently observed at the Pz site (*t* = −0.587, *p* = 0.576).

Notably, the T2 group did not show a significant suppression of N1 amplitude in the Talk condition, compared with the Listen condition, at the Fz (*t* = 1.135, *p* = 0.276), Cz (*t* = 0.328, *p* = 0.748), and Pz (*t* = 0.219, *p* = 0.830) sites, indicating an altered EC mechanism in the T2 group (Figure 1E). These results indicated that the EC mechanism modulates the auditory response according to hearing level in tinnitus patients.

### 3.3. Evidence for Altered EC Mechanism in Schizophrenia

Similar to the T2 group, significant differences in N1 amplitude between the Talk and Listen conditions were not found at Cz (*t* = −1.519, *p* = 0.163), Pz (*t* = 0.084, *p* = 0.935), and Fz (*t* = −2.244, *p* = 0.051) sites (Figure 1F). These results reflect an impaired EC mechanism in schizophrenia, which is consistent with the findings in earlier investigations.

### 3.4. Functional Connectivity Analysis for the Preparatory Phase of Self-Generated Voice vs. Listening to Playback of One’s Own Voice

In the functional connectivity of the alpha-band range (8–13 kHz) for the auditory pre-stimulus time window (−300 ms to 0 ms (the onset of self-generated vocalizations)), the dominant spatiotemporal characteristics originating from the superior temporal gyrus (STG) and the hippocampal region in the left hemisphere (Brodmann area 34L, BA34L) were observed in all groups and conditions. Because language processing is left-lateralized and hippocampal regions contribute to the language process associated with memory formation, we hypothesized that the initial observation simply reflected language processing, not the preparatory phase for auditory perception in the vocal production paradigm [33,34]. To focus on the auditory perception process, BA34 in the left hemisphere was therefore excluded.

Interestingly, Granger causality analysis showed connectivities from the right STG and hippocampal region (BA34R) to the left parahippocampal region (BA27L) and the primary auditory cortex (BA41L) for the preparatory phase of self-generated vocalization in both the healthy and T1 groups. Additional temporal connectivities toward the dorsal anterior cingulate cortex (dACC, BA32R) were present in the T1 group compared to the healthy group. Notably, the basic framework was consistently observed in the T2 group. However, the T2 group showed more complicated causal connectivity (hyperactivity) originating from the hippocampal region (BA34R) to the STG (BA22L/R), dACC (BA32L/R), auditory cortex (BA4L/R1.42L/R), Broca’s area (BA45L/R), and dorsolateral prefrontal cortex (BA46L/R); this was a distinctive feature as compared with the healthy and T1 groups (Figure 2, Appendix A).

We speculated that the connectivities from the right STG and hippocampal region (BA34R) to the left parahippocampal region (BA27L) and primary auditory cortex (BA41L) for the preparatory phase of self-generated vocalization might indicate active auditory sensing for two reasons (Figure 2A). First, those connectivities were consistently observed in the healthy and T1 groups with a similar auditory attenuation pattern, and second, they were not observed in the preparatory phase of the passive listening paradigm.

Additionally, hyperconnectivity from the hippocampal regions to the auditory cortices suggests alteration of the active auditory sensing, and that from the hippocampal regions to the frontal regions likely suggests additional cognitive loading in the T2 group. Interestingly, no connectivity was observed in the schizophrenia group for the preparatory phase of both the Talking and Listening paradigms (Figure 2C).

### 3.5. Functional Connectivity Analysis of Listening to Self-Generated Vocalization vs. Listening to Playback of One’s Own Voice

Compared with the preparatory phase for auditory perception, the actual auditory-stimulus time window (approximately 0–250 ms), reflecting the auditory perception by self-vocalization (reafferent), showed that the influences of dominant regions over other brain areas shifted from the hippocampal (BA34R) to the parahippocampal (BA27L) region in the healthy (Appendix A) and T1 groups (Figure 3). We found outflows of information from shifted regions (parahippocampal regions, BA27) to the auditory cortices and frontal regions in both the healthy and T1 groups (Figure 3). However, this shift was not observed in the T2 group, which might relate to attenuation failure of the auditory N1 response, reflecting the attentional bias (consistent hippocampal dominance, see Figure 3 and Figure 4).

The schizophrenia group had a distinctive spatiotemporal pattern in which few directional connectivities from the parahippocampal region (BA27) to dACC (BA32) were found (Figure 3C, Appendix A). The absence of causal connections before the onset of the stimulus and the few connections for auditory perception by self-vocalization might suggest extensive deficits of auditory perception in schizophrenia.

## 4. Discussion

In the Bayesian brain model, predictions are sent to lower levels, and prediction errors between prediction and sensory input are sent back to higher levels, which operate dynamically for predictive coding to readjust upcoming situations. Based on the statistical optimality principle, the Bayesian brain contributes to sensory integration as well as sensorimotor integration [1]. To predict the sensory consequences of self-produced behaviors and actions, human motor systems use neural representation from actual motor commands, which is presumably mediated by the EC mechanism [35,36,37]. In the auditory domain, functional deficits of the EC mechanism have been shown to be associated with false perceptions, such as auditory hallucination in patients with schizophrenia [4,7,38,39,40], which may result from failure of sensorimotor learning in the Bayesian forward inference. In addition, the Bayesian inference contributes to stable auditory perception in sensory integration without motor command. Auditory deafferentation may result in sensory prediction errors in the Bayesian framework. Phantom perception, such as tinnitus, can be explained by sensory integration failure in the Bayesian brain model based on evidence that tinnitus perception is entirely derived from central auditory system hyperactivity following deafferentation of the peripheral auditory system, with the frequency spectrum of tinnitus resembling the individual’s hearing frequency pattern [11,12,41], and cochlear damage primarily results in the perception of tinnitus [11,41]; additionally, prediction errors can arise because of discrepancies between the inputs predicted and delivered by auditory deafferentation in tinnitus [10,13,14].

Within the framework of the Bayesian brain model, we hypothesized that significant sensory prediction error due to peripheral deprivation (tinnitus) may lead to failure of sensorimotor integration. Therefore, we investigated the EC mechanism patterns in individual groups; healthy, tinnitus, and schizophrenia groups, in which we focused on the difference in sensory attenuation patterns to self-generate sound. As we could not determine apparent sensory prediction error, tinnitus patients were divided into two groups based on self-perceived hearing impairment. The significant subjective hearing loss group has a high threshold level for the hearing test. The tinnitus group with significant subjective hearing loss revealed high audiometric thresholds as compared to those in the tinnitus group without significant subjective hearing loss (Table 1).

Numerous investigations have explored neurobiological measures to elucidate the EC mechanism in the auditory system; attenuation of the N1 component in ERPs using the vocal production protocol is normally present, demonstrating the integrity of the EC mechanism [6,7]. The results confirmed attenuation of auditory N1 amplitude in the Talk condition (self-generated sound) compared with the Listen condition in healthy volunteers. Therefore, self-generated reafferent components (inner speech) were eliminated by intrinsic neuronal signals (ECs). Conversely, suppression of the N1 amplitude was not observed in patients with schizophrenia, consistent with previous reports [4,6,38,39]. Notably, N1 suppression in the Talk condition was observed in the T1 group but the suppression was not apparent in the T2 group, indicating that the cancellation of sensory reafference during self-generated vocalization may have failed in the T2 group with significant hearing impairment. These findings suggest that impairment of the sensory integration due to auditory sensory deprivation may influence the EC mechanism.

### 4.1. Functional Connectivity for the Preparatory Phase of the Talk and Listen Paradigm

Here, we further investigated the underlying spatiotemporal evidence for auditory processing during the Talk and Listen paradigm to clarify distinctive features between phantom perception and false perception in the auditory domain. In the present study, the Granger causal connectivity method at the grand-averaged evoked alpha activity of the frequency range between 8 and 13 Hz from −300 ms to 0 ms pre-stimulus for investigating the early auditory attention process was used. Then it was investigated from 0 ms to 250 ms post-stimulus for exploring the auditory response (possible substrates for auditory attenuation) by self-voice reafferent and by the playback of the participant’s own voice. Although inverse problems from the scalp EEG to cortical source distribution may exist, this approach has been validated as a reasonable method for causal connectivity analysis [27,28].

As mentioned above, the focus in the present study was on the spatiotemporal patterns of auditory processing, rather than language processing. Thus, the hippocampal region (BA34) in the left hemisphere was excluded from the final spatiotemporal connectivity analysis because language processing is left-lateralized and the left hippocampal region is linked to the language process associated with memory formation [33,34]. Furthermore, the connectivity matrix for our initial ROI contained a large number of connections from BA34L, making the results very hard to interpret. Thus, we reduced the connectivity network by taking out the ROI (BA34) and found that it did not significantly affect the remaining connectivity overflow when the ROI was removed (Appendix A). After subtracting the ROI, Granger causality analysis revealed that the right STG and the hippocampal region (BA34R) were the dominant regions communicating to the parahippocampal region (BA27L) and primary auditory cortex (BA41L) for the preparatory phase of vocal production (Talk paradigm) in the healthy group; we speculated that the outflows would indicate the potential pathway of the EC mechanism in the auditory domain since (1) the outflows from hippocampal regions were transmitted to the auditory cortex before the actual sound was not delivered to the inner ear (Figure 2 and Figure 4), and (2), and the outflows were not found in all groups during the Listen paradigm.

Interestingly, even though the healthy and T1 groups showed a similar causal connectivity pattern, additional connectivity from the hippocampus (BA34) to the dACC (BA32) was found in the T2 group. There is increasing evidence that the hippocampus aids in processing the (mis)match between the expected sensory consequences of speaking and the perceived speech feedback [42,43,44]. Furthermore, animal studies have demonstrated that the STG and middle temporal gyrus respond more to self-generated speech than to other-generated speech [45,46]. Considering the auditory attenuations to the self-generated voice were equally present in both the healthy and T1 groups, the directional connectivity from STG and the hippocampal region (BA34) to the auditory cortex indicates plausible evidence for normal auditory active sensing.

Notably, previous investigations using functional magnetic resonance imaging showed increased activity in the auditory cortex and STG, lateral and medial prefrontal cortex, and premotor areas under impaired speech feedback [47,48]. Accordingly, we speculated that hyperconnectivity from the STG to the hippocampal region, frontal cortex, and auditory cortex might be responsible for the failure of auditory attenuation in ERPs, indicating the presence of aberrant signals in the T2 group. In the schizophrenia group, those dominant connectivities were absent, which indicates that the EC mechanism during pre-attentive phases of information processing is extensively damaged in affected patients [40,49,50]. Thus, although auditory N1 attenuation failure was observed in both the T2 and schizophrenia groups, the underlying neural signature resulting in N1 suppression failure differed between the two groups.

### 4.2. Functional Connectivity for Auditory Perception in the Talk and Listen Paradigm

Connections for auditory perception by reafferent sound in the Talk paradigm showed a dominance shift (Figure 3 and Figure 4), in which the contralateral parahippocampal region (BA27) in the healthy group, and the hippocampal and parahippocampal regions (BA34 and BA27, respectively) in the T1 group, had primary influence over other brain areas. However, in the T2 group, this dominance shift was not observed; the hippocampal area was consistently dominant. Therefore, we speculated that outflows toward auditory cortices originated from the contralateral parahippocampal region (left side, BA27) might be attenuated by delivered signals from STG and the hippocampal region (BA34) before vocalization; this might explain the resultant N1 attenuation in ERPs because the shift was partly present in the T1 group but not in the T2 group.

Additionally, the causal connectivities of the dorsolateral prefrontal region (BA46), dACC (BA32), and inferior frontal region (BA45) originating from the contralateral parahippocampal region (left side, BA27) were consistently observed in both the healthy and T1 groups. Because language functions are strongly lateralized to the left side of the brain [34], we postulated that both hemispheric connections between the hippocampal/parahippocampal regions would respond to monitor sensory consequences; moreover, we expected that the frontal region would function as a comparator (mis)matching between actual sensory feedback versus predicted sensory feedback. However, in this study, the T2 group had deficits in those connectivities. Notably, in the schizophrenia group, the few connections that formed during actual stimulation of sound reafference by self-vocalization might be responsible for the dysfunctional EC mechanism.

Taken together, our findings indicate that the functional deficits of the EC mechanism were presented in tinnitus patients with significant hearing impairment, indicating that phantom perception after sensory deprivation might lead to impairment in EC mechanism. However, the corresponding neural representation with spatiotemporal patterns presented differently between groups. Thus, sensory attenuation failure in tinnitus with significant hearing impairment may have resulted from aberrant signals, not fundamental deficits in EC mechanism as for an auditory hallucination.

## 5. Conclusions

In summary, functional impairment of the EC mechanism underlies the phenomenological aspects of phantom perception due to substantial sensory prediction errors following sensory deprivation in the auditory domain. However, the spatiotemporal signature responsible for the impairment of the EC mechanism differed between false perception and phantom perception. The present study had several constraints, such as concerns regarding the experimental design such as small sample size and subjective allocation, and methodological limitations and technical limitations such as EEG source localization [51,52] and noise filtering from muscle movements by the voice. In particular, since the N1 suppression effects might be influenced by the tinnitus masking level or hearing loss in the T2 group, this study should be substantiated with further investigation of hearing loss subjects without tinnitus to clarify our results regarding the alteration of the EC mechanism in tinnitus. However, current results might provide a new perspective on neurobiological aspects of abnormal auditory perception resulting from deficits in predictive coding.

## Figures and Tables

**Figure 1 biology-11-01501-f001:**
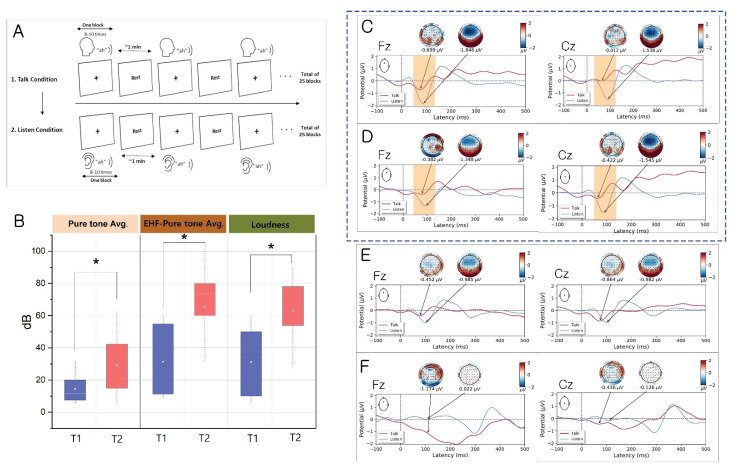
ERP results from Talk and Listen paradigm protocol and group characteristics. (**A**) Illustration of the Talk/Listen paradigm. The paradigm was composed of two tasks (Talk and Listen) performed consecutively. Each task consisted of 25 blocks. In the first task (Talk condition), participants produced the sound “ah” 8 to 10 times. The “ah” utterances were recorded for all blocks; participants then listened to their own recorded utterances during the Listen condition. A “Rest” sign appeared upon the completion of each block, allowing the participant to rest between blocks. (**B**) Representative graph for the T1 and T2 groups. Data are shown as mean ± SD. * indicates statistical significance (*p* < 0.05). (**C**–**F**) Grand-averaged ERP time courses in the Talk condition (solid red line) and Listen condition (solid blue line) for the healthy controls (**C**), T1 (**D**), T2 (**E**), and schizophrenia groups (**F**) at the Fz and Cz electrodes and the topographies of their maxima. Note the significant attenuation of auditory N1 amplitude during the Talk condition, compared with the Listen condition, in the healthy control (**C**) and T1 groups (**D**). Color bar indicates amplitude (µV). All topographies are shown from the vertex view with the nose at the top.

**Figure 2 biology-11-01501-f002:**
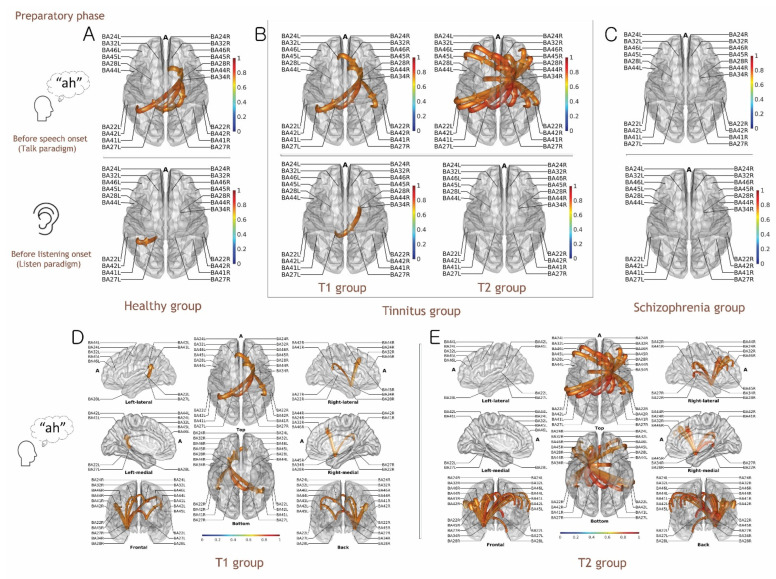
Granger causal connectivity of grand-averaged evoked alpha activity during an auditory pre-stimulus time window (−300 ms to 0 ms) in the Talk and Listen paradigm. (**A**) Healthy group. (**B**) Tinnitus groups: T1 (tinnitus without subjective hearing loss), T2 (tinnitus with subjective hearing loss). (**C**) Schizophrenia group. (**D**) Spatial representation of connectivity in the T1 group. (**E**) Spatial representation of connectivity in the T2 group. Color-scaled directional arrows link two causally connected regions of interest (ROIs) across 21 ROIs (BA24L/R, BA22L/R, BA27L/R, BA28L/R, BA32L/R, BA34R, BA41L/R, BA42L/R, BA44L/R, BA45L/R, and BA46L/R). Note that during the auditory pre-stimulus phase (before vocalization) of the Talk paradigm, outflows of information from the right hippocampal regions (BA34R) to the left auditory cortex (BA41L) were observed in the healthy group, suggesting a potential pathway of efference copy signals, which were found consistently in the T1 group. Notably, the outflows of information were more prominent in the T2 than in the T1 group, and additional outflows to the frontal regions were found in the T2 group. Interestingly, those functional connectivities were not observed in the schizophrenia group. (A: anterior).

**Figure 3 biology-11-01501-f003:**
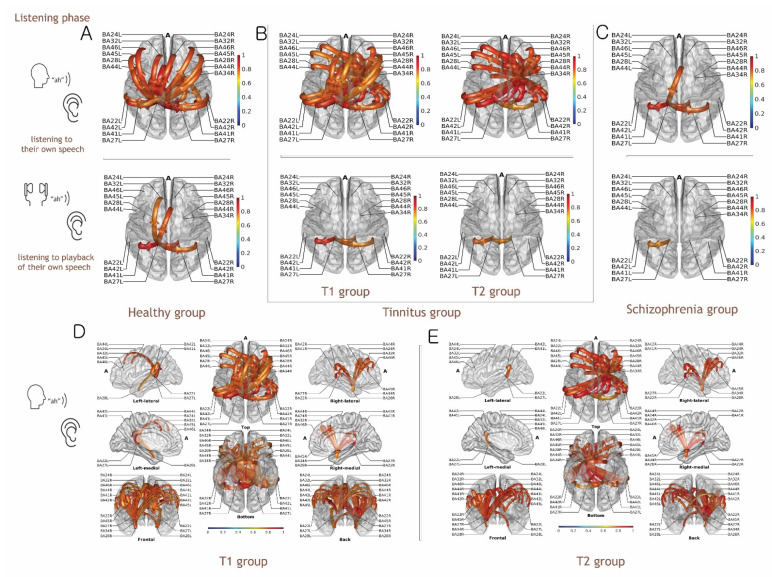
Granger causal connectivity of grand-averaged evoked alpha activity during auditory stimulus time window (0 to 200 ms) in the Talk and Listen paradigm. (**A**) Healthy group. (**B**) Tinnitus groups: T1 (tinnitus without subjective hearing loss), T2 (tinnitus with subjective hearing loss). (**C**) Schizophrenia group. (**D**) Spatial representation of connectivity in the T1 group. (**E**) Spatial representation of connectivity in the T2 group. During the Talk paradigm, a dominance shift from hippocampal regions (BA34R) to left parahippocampal regions (BA27L) was observed in the healthy and T1 groups, compared with preparatory phase. However, the dominance shift was not observed in the T2 group. A few connectivities were present in the schizophrenia group. Color-scaled directional arrows link two causally connected regions of interest (ROIs) across 21 ROIs (BA24L/R, BA22L/R, BA27L/R, BA28L/R, BA32L/R, BA34R, BA41L/R, BA42L/R, BA44 L/R, BA45L/R, and BA46L/R). (A: anterior).

**Figure 4 biology-11-01501-f004:**
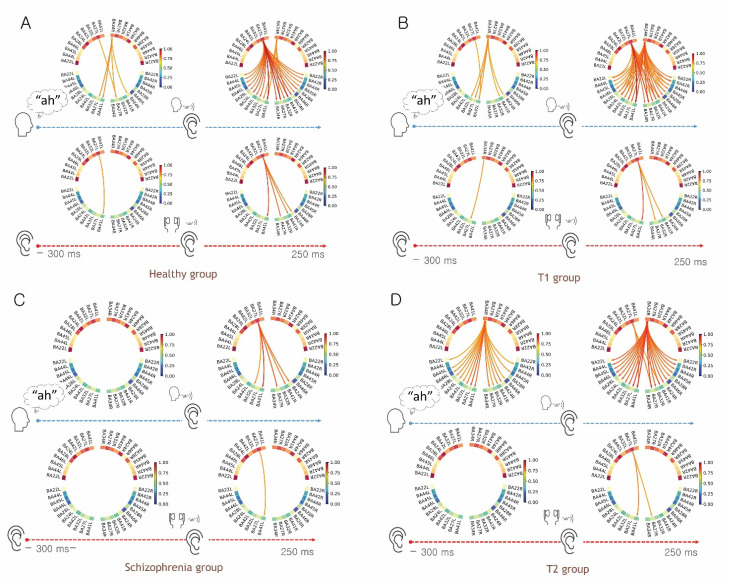
Representation of dynamic change in causal connectivity for grand-averaged evoked alpha activity. (**A**) Healthy control group. (**B**) T1 group (tinnitus without subjective hearing loss). (**C**) Schizophrenia group. (**D**) T2 group (tinnitus with subjective hearing loss). Note that reafferent auditory stimulation led to a dominance shift during the Talk paradigm in healthy control and T1 groups. However, the dominance shift was not observed in the T2 group. Outflows were not observed in the schizophrenia group. Color-scaled directional arrows link two causally connected regions of interest (ROIs) across 21 ROIs (BA24L/R, BA22L/R, BA27L/R, BA28L/R, BA32L/R, BA34R, BA41L/R, BA42L/R, BA44 L/R, BA45L/R, and BA46L/R).

**Table 1 biology-11-01501-t001:** Demographics of the study participants.

Variables	Tinnitus		Schizophrenia	Controls
	T1	T2		
Total Number (*n*)	8	15	10	23
Gender (female/male)	2 /6	6/9	7/3	12/11
Age ± SD	39.12 ± 12.59	40.40 ± 13.50	30.10 ± 6.00	34.76 ± 7.40
Pure tone average * (dB) for the right Ear	12.03 ± 7.70	27.33 ± 20.11	7.87 ± 9.82	5.27 ± 3.40
High frequency pure tone average ** (dB) for the right ear	29.68 ± 19.93	57.94 ± 28.85		
Pure tone average (dB) for the left ear *	11.56 ± 9.08	20.25 ± 15.81	4.12 ± 2.63	4.13 ± 3.90
High frequency pure tone average (dB) for the left ear	23.85 ± 21.57	46.94 ± 13.50		
Pure tone average (dB) for the involved side *	14.50 ± 10.08	29.10 ± 19.16		
High frequency pure tone average (dB) for the involved side	31.35 ± 22.07	65.38 ± 26.6		
Tinnitus side (*n*)				
Right/Left/Both	1/3/4	10/2/3		
Pitch in kHz	6.00 ± 2.83	5.25 ± 2.98		
Loudness in dB HL	31.28 ± 20.46	63.08 ± 20.35		
THI score	57.00 ± 25.16	42.93 ± 18.39		
PANSS score			52.70 ± 14.23	
Hallucinatory behavior score on PANSS			2.30 ± 1.25	

SD indicates standard deviation, PTA indicates pure tone average, THI indicates tinnitus handicap inventory, and PANSS indicates positive and negative syndrome scale. * Average of hearing threshold levels at 0.5 kHz, 1 kHz, 2 kHz, and 3 kHz; ** Average of hearing threshold levels at 8 kHz and 12 kHz.

## Data Availability

The datasets, including connectivity matrices, are available from the corresponding author on reasonable request.

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
