# Peer review of "Neurobiological Signatures of Auditory False Perception and Phantom Perception as a Consequence of Sensory Prediction Errors"

_biology, 2022, doi:10.3390/biology11101501_

Round 1

Reviewer 1 Report

This article is very well organized.

It seems to me that the statistical methods are well applied.

The audiological part very well elaborated and clear, including the Analysis of auditory-ERPs.

Maybe you could rearrange the figures so that they are clearer, because the numerical part becomes excessively small, in order to have better readability.

I leave to the consideration of someone more specialized in the neurological field, some more detailed evaluation.

In view of this, I have nothing to contradict the publication of this article.

Author Response

We sincerely appreciate your review, and as you recommended, we revised the lower cases to capital letters in the Figure's alphabetical order. In addition, we changed our figures with high resolution;  please see the revised MS

We believe that the manuscript has been significantly improved, thanks to the reviewers’ valuable comments. We hope that this revised manuscript is acceptable for publication

Thank you   

Reviewer 2 Report

The authors used electroencephalography and Granger causality analysis to explore EC mechanisms in 23 healthy controls, 23 patients with tinnitus (accompanied by mild (or substantial hearing loss), and 10 patients with schizophrenia who experienced auditory hallucinations. Auditory responses to self-generated sound in a “Talk/Listen” paradigm were suppressed in healthy controls and in the tinnitus/mild hearing loss group, but less so in a group with tinnitus/substantial hearing loss. The group with schizophrenia differed from all other groups in interesting ways. The study offers an interesting look at predictive coding and EC feedback processes, with potentially important implications for understanding auditory hallucinations in schizophrenia, and perhaps phantom sound perception (tinnitus) as well.

I have one major concern regarding the study design and its potential influence on the results, namely, the use of a single consistent playback level of 75-85 dB SPL for all groups. Because the playback level was fixed, the effective stimulus level was higher for the control group than for either of the tinnitus groups (substantially so for the T2 group). The differences between the control group and the patients with schizophrenia are not accounted for by a difference in stimulus level, but it seems possible that differences between the healthy controls and the tinnitus groups could be explained at least in part by differences in stimulus level. The authors should address this potential confound of using different effective stimulus levels for the normal hearing, mild hearing loss, and substantial hearing loss groups.

--The suppression effect appears smaller for the T1 group vs. healthy controls—might this reflect the lower effective level of the stimulus in the presence of masking by their tinnitus? Similarly, might the much smaller suppression effect seen in the T2 group be accounted for by different effective level of the stimulus? The T2 participants were hearing the stimuli at a lower HL in the Listen condition due to their hearing loss; would the suppression effect be present if hearing levels were equated?

--In the control group, if masking were used to simulate tinnitus, would the results parallel those of T groups? Conversely, if the level of the stimulus is reduced for controls, would their results match the results from the hearing impaired group?

--Short of performing additional studies on control participants or tinnitus patients, a simple correlation or regression analysis might address this. Is there an inverse predictive relationship between degree of hearing loss and magnitude of suppression?

Other:

--p. 3, Please modify the statement about using high-frequency audiometry to “investigate the possibility of hidden hearing loss,” as HFA does not reveal hidden hearing loss.

--Knowing the distribution of VAS grades for patients in the T2 group might be helpful

--p. 3 and elsewhere, provide full statistics (test statistic, df, value, p, and effect size (not dependent on sample size)), not simply a p value (or a t value without df)

--p. 3, procedures for assessing tinnitus pitch and loudness not described

--p. 6, specifically, how was the N1 suppression effect estimated? Are you referring to a simple difference score (amplitude in Talk condition minus Listen conditions)?

--p. 6. “Baseline corrections were performed…”—when was this done? (This sentence likely should come earlier in the description of EEG data analysis)

--p. 8, language processing is left-lateralized in most, not all (about 95% of right-handed participants). Was handedness of the participants assessed? Was there evidence from individual data that all participants had language lateralized to their left hemisphere?

Author Response

The reviewers’ comments on the manuscript were constructive, and we have addressed each point they raised in this revision. Please see attached for point by point response.  We believe that the manuscript has been significantly improved, thanks to the reviewers’ valuable comments. We hope that this revised manuscript is acceptable for publication

Reviewer 3 Report

This study investigated the efference copy (EC) mechanism in the tinnitus model with the hypothesis that the EC mechanism would be impacted in tinnitus patients with self-reported hearing deficits. EEG recordings were performed in response to a Talk/ Listen protocol on four groups of participants: controls, tinnitus with hearing impairment, tinnitus without hearing impairment and individuals with schizophrenia.  

This is a well-designed study and the methods are described in detail. I just have a few comments as below.

-          -The authors mention that participants with tinnitus who failed pitch matching were excluded. Please explain why this was decided.

-          -The authors mention that “self-perceived hearing impairments were correlated with objective hearing tests, including the high-frequency range (Table 1).”  I don’t see any correlation findings reported in Table 1 but it would help to support this statement if these were added.

-          -Table 1 reports “loudness in dB HL”. Is that the perceived loudness of tinnitus? If so, how was it measured?

-          -The methods mention “interpolation of channels that were not functioning appropriately”. What methods were used for the interpolation?

-          -EEG recordings from Fz, Cz and Pz were considered. Could the authors add why these three locations were selected?  

-          -The sentences at the start of 3.4 would be well-placed in the methods section (2.5) to summarise what the authors aimed to do:

“In the present study, the functional connectivity of the alpha-band range (8–13 kHz) for the auditory prestimulus time window (-300 ms to 0 ms [the onset of self-generated vocalizations]) was investigated for the early attentional networks for auditory perception.”

-        -The section below is unclear and needs proof-reading:

 “these findings indicated the potential pathway of EC mechanism in the auditory domain as information outflows from hippocampal regions transmitted to the auditory cortex even the actual sound was not delivered to the inner ear (Figs 2 and 4). In contrast, the outflows did not find in all groups during listen paradigm”.

-         -The section explaining that “attenuation of the N1 component in ERPs using the vocal production protocol is normally present” would benefit from being mentioned earlier (in the Introduction) since this concept forms the basis of a big part of the results.

-       -The ‘Simple Summary’ section uses terms such as Bayesian inference, efference copy and N1 attenuation. These need to be briefly described, to meet the purpose of a simple summary.

Author Response

(The authors gave the same response as above.)

Round 2

Reviewer 2 Report

The study and results are interesting, but I still have an issue with the main interpretation of the results as showing that the EC mechanism is altered in the T2 group, and that the alteration is related to tinnitus perception. 

The authors have not provided convincing evidence for that the EC mechanism is actually altered in the T2 group. The results might be accounted for simply on the basis of reduced stimulus levels, for both Talk and Listen conditions. “Ah” sounds were produced at a calibrated level of about 75-85 dB, and the same level was used in the playback Listen condition. Control and T1 participants heard the Ah sounds, both self generated and played back, at similar levels (around 75 dB SPL), while the T2 participants heard them at substantially reduced levels (reduced re: normal hearing by about 20 dB SPL). The suppression effect (difference between Talk and Listen conditions) could be reduced not because of an altered EC mechanism but simply because the stimulus level was substantially lower (by at least 20 dB) for the T2 participants due to their hearing loss. Until the authors rule out differences in effective stimulus level as an explanation for the differences in suppression seen in the T2 hearing loss group and controls, the conclusion that the EC mechanism is altered is premature; it could be what normally happens when lower levels of sound (55 dB SPL) are generated and heard.

Also, changes are needed in the Abstract::

1) first sentence and last sentence, do you mean auditory differentiation or auditory deafferentation? If auditory differentiation is correct, then explain the term as this is not a common term and its meaning is unknown.

2) why “apparent hearing impairment”? Hearing was clinically assessed, so the hearing impairment is documented; “apparent” implies that impairment is assumed but not directly known

Other grammar changes are needed throughout the paper.

Author Response

We sincerely appreciate the reviewer's comments. we revised our MS according to the concern that the reviewer raised.  

We realized that our conclusion would be premature because we did not include hearing loss subjects without tinnitus.  Thus, we toned down our conclusion and removed Figure 5 in our conclusion. 

Please find attached our response .

---------------------------------------

The study and results are interesting, but I still have an issue with the main interpretation of the results as showing that the EC mechanism is altered in the T2 group, and that the alteration is related to tinnitus perception. 

The authors have not provided convincing evidence for that the EC mechanism is actually altered in the T2 group. The results might be accounted for simply on the basis of reduced stimulus levels, for both Talk and Listen conditions. "Ah" sounds were produced at a calibrated level of about 75-85 dB, and the same level was used in the playback Listen condition. Control and T1 participants heard the Ah sounds, both self generated and played back, at similar levels (around 75 dB SPL), while the T2 participants heard them at substantially reduced levels (reduced re: normal hearing by about 20 dB SPL). The suppression effect (difference between Talk and Listen conditions) could be reduced not because of an altered EC mechanism but simply because the stimulus level was substantially lower (by at least 20 dB) for the T2 participants due to their hearing loss. Until the authors rule out differences in effective stimulus level as an explanation for the differences in suppression seen in the T2 hearing loss group and controls, the conclusion that the EC mechanism is altered is premature; it could be what normally happens when lower levels of sound (55 dB SPL) are generated and heard.

Response) We fully understood your concern and realized our conclusion can be misunderstood by making rash conclusions. Thus, we revised our abstract conclusion according to the reviewer's comments.

We added the sentence regarding limitations in abstract

Even though the present study had significant implicational constraints in that we did not include hearing loss subjects without tinnitus, these results might suggest that auditory deafferentation (hearing loss) may result in sensorimotor integration using EC signals   

Also, we removed Fig 5 and added the limitation in the conclusion

In particular, since the N1 suppression effects might be influenced by tinnitus masking level or hearing loss in the T 2 group, this study should be substantiated with further investigation of hearing loss subjects without tinnitus to clarify our results regarding alteration of EC mechanism in the tinnitus

Also, changes are needed in the Abstract::

  • first sentence and last sentence, do you mean auditory differentiation or auditory deafferentation? If auditory differentiation is correct, then explain the term as this is not a common term and its meaning is unknown

Response) This was a typo. We corrected the differentiation to deafferentation and added the word for the meaning of hearing loss. Thank you for your meticulous review.    

  • why "apparent hearing impairment"? Hearing was clinically assessed, so the hearing impairment is documented; "apparent" implies that impairment is assumed but not directly known

Response) As all tinnitus underlie hearing loss in an audiogram, we used the word "apparent" to differentiate mild vs. significant in our tinnitus group.

We revised the word- "apparent" to "significant." 

Other grammar changes are needed throughout the paper.

Response) The English in this document has been checked twice by two professional editors, both native speakers of English.

For a certificate, please see: http://www.textcheck.com/certificate/Yy1WLJ

-------------------------------------------

Thank you again 

Round 3

Reviewer 2 Report

Adequate revisions have been made to the manscript.